# Association between *ABCB1* rs2235048 Polymorphism and THC Pharmacokinetics and Subjective Effects following Smoked Cannabis in Young Adults

**DOI:** 10.3390/brainsci12091189

**Published:** 2022-09-03

**Authors:** Justin Matheson, Yollanda J. Zhang, Bruna Brands, Christine M. Wickens, Arun K. Tiwari, Clement C. Zai, James L. Kennedy, Bernard Le Foll

**Affiliations:** 1Translational Addiction Research Laboratory, Centre for Addiction and Mental Health, University of Toronto, 33 Ursula Franklin Street, Toronto, ON M5S 2S1, Canada; 2Department of Pharmacology and Toxicology, University of Toronto, Toronto, ON M5S 1A8, Canada; 3Institute for Mental Health Policy Research, Centre for Addiction and Mental Health, Toronto, ON M5S 3M1, Canada; 4Controlled Substances and Cannabis Branch, Health Canada, Ottawa, ON K1A 0K9, Canada; 5Dalla Lana School of Public Health, University of Toronto, Toronto, ON M5T 3M7, Canada; 6Campbell Family Mental Health Research Institute, Centre for Addiction and Mental Health, Toronto, ON M5T 1R8, Canada; 7Institute of Health Policy, Management and Evaluation, University of Toronto, Toronto, ON M5T 3M6, Canada; 8Neurogenetics Section, Tanenbaum Centre for Pharmacogenetics, Molecular Brain Science, Campbell Family Mental Health Research Institute, CAMH, Toronto, ON M5T 1R8, Canada; 9Department of Psychiatry, University of Toronto, Toronto, ON M5T 1R8, Canada; 10Institute of Medical Science, University of Toronto, Toronto, ON M5S 1A8, Canada; 11Laboratory Medicine and Pathobiology, University of Toronto, Toronto, ON M5S 1A8, Canada; 12Broad Institute, Cambridge, MA 02142, USA; 13Acute Care Program, Centre for Addiction and Mental Health, Toronto, ON M6J 1H4, Canada; 14Department of Family and Community Medicine, University of Toronto, Toronto, ON M5G 1V7, Canada; 15Waypoint Research Institute, Waypoint Centre for Mental Health Care, Penetanguishene, ON L9M 1G3, Canada

**Keywords:** cannabis, THC, *ABCB1*, P-glycoprotein, pharmacogenetics

## Abstract

Genetic influences on acute responses to psychoactive drugs may contribute to individual variability in addiction risk. *ABCB1* is a human gene that encodes P-glycoprotein, an ATP-dependent efflux pump that may influence the pharmacokinetics of delta-9-tetrahydrocannabinol (THC), the primary psychoactive component of cannabis. Using data from 48 young adults (aged 19–25 years) reporting 1–4 days of cannabis use per week who completed a placebo-controlled human laboratory experiment, we tested the hypothesis that the rs2235048 polymorphism of *ABCB1* would influence acute responses to smoked cannabis. C-allele carriers reported on average greater frequency of weekly cannabis use compared to the TT genotype carriers (TC/CC mean ± SEM = 2.74 ± 0.14, TT = 1.85 ± 0.24, *p* = 0.004). After smoking a single cannabis cigarette to their desired high, C-allele carriers had higher area-under-the-curve (AUC) of both THC metabolites (11-OH-THC TC/CC = 7.18 ± 9.64, TT = 3.28 ± 3.40, *p* = 0.05; THC-COOH TC/CC = 95.21 ± 116.12, TT = 45.92 ± 42.38, *p* = 0.043), and these results were impact by self-reported ethnicity. There were no significant differences in self-reported subjective drug effects except for a greater AUC of visual analogue scale rating of drug liking (TC/CC = 35,398.33 ± 37,233.72, TT = 15,895.56 ± 13,200.68, *p* = 0.017). Our preliminary findings suggest that further work in a larger sample should investigate whether human *ABCB1* influences cannabis-related phenotypes and plays a role in the risk of developing a cannabis use disorder.

## 1. Introduction

Cannabis is one of the most widely used psychoactive drugs worldwide [1]. Recent data from the United Nations Office on Drugs and Crime (UNODC) estimates that 4% of the global population used cannabis at least once in 2019, representing 200 million individuals [2]. Acute effects of cannabis include both positive and negative subjective effects, dose-dependent impairment of cognitive skills, and transient perceptual alterations and psychotomimetic effects [3]. Some individuals who use cannabis regularly will go on to develop a cannabis use disorder (CUD), a diagnosis characterized by a problematic pattern of cannabis use that leads to clinically significant impairment and distress [1,4]. Given the large number of individuals using cannabis globally, understanding the causes of individual variation in CUD susceptibility is of central importance to public health.

Genetics are thought to play a role in determining susceptibility to a variety of cannabis-related phenotypes, including CUD [5]. A meta-analysis of 24 twin studies estimated the heritability of problematic cannabis use to be 51% for males and 59% for females [6]. Genome-wide association studies (GWAS) have identified some variants associated with cannabis-related phenotypes [5], but often the phenotype of interest (e.g., lifetime cannabis use or lifetime CUD) is so broad, it is difficult to determine what role these genes play in determining cannabis use or harms. Laboratory-based candidate gene studies can help to fill this gap by examining associations between well-defined cannabis-related phenotypes (e.g., subjective responses to acute cannabis exposure) and genetic polymorphisms in genes of interest. For example, we have previously identified greater subjective responses to smoked cannabis in carriers of the rs1049353 T-allele and rs2023239 C-allele of the cannabinoid receptor 1 (*CNR1*) gene [7] and the rs510769 T-allele of the mu-opioid receptor gene (*OPRM1*) [8].

ATP-binding cassette sub-family B member 1 (ABCB1), also known as multidrug resistance protein 1 (MDR1) or P-glycoprotein 1 (P-gp), is an ATP-dependent efflux pump encoded by the human *ABCB1* gene, which was first identified in 1976 as a transporter that modulates drug permeability and resistance [9]. In the brain, expression of P-gp on endothelial cells of the blood–brain barrier (BBB) limits distribution of a wide variety of drugs into the central nervous system (CNS) [10]. Thus, P-gp can play a significant role in the pharmacokinetics of a drug (i.e., absorption into the CNS, distribution to the active site, and excretion), which in turn can influence its pharmacodynamic effects [8]. A number of single-nucleotide polymorphisms (SNPs) have been identified in the *ABCB1* gene that have been associated with drug pharmacokinetics or pharmacodynamics, including the synonymous C to T exchange at nucleotide 3435 in exon 26 (C3435T, rs1045642) that is thought to potentially alter the tertiary structure of the protein and thus its stability or expression [11]. The C3435T polymorphism has been found to impact absorption and tissue concentrations of a number of P-gp substrates. The T allele was associated with lower duodenal expression of P-gp and higher plasma levels of the P-gp substrate digoxin in healthy volunteers [12]. A more recent study found the T allele was associated with a greater brain/blood methadone concentration ratio, suggesting greater brain concentrations of methadone in T-allele carriers [13].

Converging lines of evidence have suggested a role of *ABCB1* variation in cannabis-related phenotypes. In their 2008 study, Bonhomme–Faivre and colleagues found that P-gp deficient mice had significantly higher total exposure to THC (area under the plasma concentration time curve) than wild-type mice after exposure to oral THC, suggesting that THC is a substrate of P-gp [14]. Subsequently, this same group found a significantly higher frequency of the C allele and CC genotype of the *ABCB1* rs1045642 SNP in patients with cannabis dependence compared to a control group [15]. Next, in a sample of patients with cannabis dependence who reported “heavy” use of cannabis (≥7 joints per week), they found a significantly higher mean plasma THC concentration in the *ABCB1* rs1045642 CC genotype group compared to T-allele carriers (TT + TC) [16]. The authors proposed a potential pharmacokinetic hypothesis of CUD susceptibility, where variability in *ABCB1* activity could lead to rapid elimination of THC from the brain, which could encourage individuals to use cannabis more frequently, at higher doses, or at shortened intervals to continue experiencing positive subjective effects or avoid negative effects such as withdrawal [15,16].

The rs2235048 SNP is an intronic variant of *ABCB1*, located 134 base pairs downstream of rs1045642. Due to their proximity in location, a near-complete linkage disequilibrium has been observed between the two SNPs, where the rs1045642 T allele is linked to rs2235048 C allele [17]. In addition, rs2235048 has been used as a proxy SNP for rs1045642 in genotyping [18]. Therefore, the findings of lower plasma THC concentration and lower cannabis dependence risk associated with rs1045642 T allele are also expected to associate with rs2235048 C allele. Based on results from previous studies, we hypothesize that the rs2235048 C allele will be associated with lower blood THC and metabolite concentrations after smoking cannabis in the human laboratory, as well as lower subjective responses to smoked cannabis.

## 2. Materials and Methods

### 2.1. Study Population

Healthy male and female adults (aged 19–25 years) were recruited in Toronto between July 2012 and August 2016. Participants used cannabis regularly (defined as 1–4 days of use per week), with evidence of recent cannabis use at eligibility screening (urine THC-COOH point-of-care cut-off of 50 ng/mL or laboratory assay cut-off of 15 ng/mL). Participants were excluded if they used alcohol on any study day (confirmed by breath alcohol concentration), used medications that affect cognition, had severe medical or psychiatric conditions, had first degree relatives diagnosed with schizophrenia, used psychoactive substances other than cannabis during the study, met the criteria for DSM-IV cannabis dependence or other substance dependence, were pregnant, trying to become pregnant, or breastfeeding.

### 2.2. Study Design and Procedure

Data were obtained from a double-blind, placebo-controlled, parallel-groups trial conducted at the Centre of Addiction and Mental Health (CAMH) in Toronto, Ontario, Canada [19]. In the parent study (the CADRI study), 90 participants were randomized using a 2:1 allocation ratio to receive active (*n* = 60) or placebo (*n* = 30) cannabis, and the effects on simulated driving performance as well as other cognitive and subjective effects were assessed [20]. During eligibility screening, participants had the option to provide an additional 20 mL blood samples to be used for DNA extraction and genetic analysis. Only participants who provided consent to this procedure in the active arm of the CADRI study (*n* = 52) were included in the current analysis. The genotypes of each participant were determined using the Infinium Global Screening Array (Illumina, Inc., San Diego, CA, USA) at the CAMH Biobank and Molecular Core Facility. The subjects were grouped based on alleles, and outcome data were analyzed between groups. More details can be found in our previous studies of *CNR1* genotype [7] and *OPRM1* genotype [8].

Following eligibility screening, eligible participants returned to CAMH for four sessions on consecutive days, including a practice session (for participants to become familiar with the testing procedures), an acute drug exposure session (administration of cannabis or placebo), and two sessions at 24 h and 48 h following initial exposure. Breathalyzer (AlertTM J5 model, Alcohol Countermeasure Systems) was used to confirm a breath alcohol content of zero before each session. Urine samples were collected for toxicology screening (QuickscreenTM CLIA-Waived 10-Panel Multi Drug Test) to confirm abstinence from drugs that were not medically required 48 h prior to the first session and for the duration of the study. Negative urine pregnancy tests were confirmed for female participants. Questionnaires were used to obtain demographic information during the practice session.

### 2.3. Intervention

During the acute drug exposure session, the participants each received one single cannabis cigarette with a mass of approximately 750 mg. Active cannabis (12.5% THC) was obtained from Prairie Plant Systems Inc., and placebo cannabis (<0.01% THC) was obtained from the National Institute on Drug Abuse (NIDA) in the United States. The cannabidiol (CBD) and cannabinol (CBN) content in both the active and placebo cannabis were <0.5%, thus considered negligible. Participants were instructed to smoke the cigarette *ad libitum* (i.e., to the participant’s desired “high”) over a maximum of 10 min in a dedicated reverse-airflow smoking laboratory. The duration of smoking was recorded, and the end of smoking was marked as time 0. The masses of the cigarette before and after smoking were measured to estimate THC dose. Estimated THC dose was calculated as the change in mass of the cannabis cigarette multiplied by the THC potency of the cannabis plant material (12.5%, i.e., 0.125).

### 2.4. Outcomes

Blood samples were collected by a registered nurse using an indwelling intravenous catheter at baseline (prior to cannabis exposure) and then at 5, 15, 30, 60, 120, 180, 240, 300, and 360 min after exposure. THC and two of its primary metabolites (11-OH-THC and THC-COOH) were measured in whole blood. Samples were first purified by solid phase extraction, derived by gas chromatography, and then analyzed by mass spectrometry. Quantification limits (LOQ) for THC, 11-OH-THC, and THC-COOH were 0.5, 1.0, and 1.0 ng/mL, respectively. We note that whole-blood cannabinoid concentrations are typically about half of the concentration found in plasma [21,22].

Subjective effects measures were collected at the same time points as blood draws. The seven visual analogue scale (VAS) items used in the study included “I feel this effect” (i.e., “Effect”), “I feel this high” (i.e., “High”), “I feel the good effects” (i.e., “Good Effects”), “I feel the bad effects” (i.e., “Bad Effects”), “I like cannabis” (i.e., “Liking”), “I feel the rush” (i.e., “Rush”), and “this feels like cannabis” (i.e., “Like Cannabis”). The participants were asked to indicate their feelings on a 100 mm unipolar line ranging from 0 (“not at all”) to 100 (“extremely”).

### 2.5. Data Analysis

Statistical analyses were performed using SPSS 27.0 for Windows. We grouped CC and TC genotype carriers together and compared to TT genotype carriers to mirror the grouping of genotypes in Kebir et al. (2018) [16]. Samples characteristics were compared between genotype groups using either an independent-samples t-test for continuous measures (age, BMI, cannabis use frequency) or a chi-squared test for categorical measures (sex, ethnicity).

A similar approach was taken for both blood cannabinoid data and VAS data: first, we ran a split-plot analysis of variance (ANOVA), where genotype was the between-subjects term and time was the within-subjects term, to determine if there was a significant interaction between genotype and blood cannabinoids or subjective effects over the course of 360 min following cannabis exposure. Whenever Mauchly’s Test of Sphericity indicated a lack of sphericity, the Greenhouse-Geisser correction is reported. Next, we ran independent samples t-tests to determine if there was a significant genotype difference in maximum values or area-under-the-curve (AUC, i.e., total exposure). AUC_0–360min_ was calculated using the trapezoidal method. Finally, we conducted a secondary analysis in the subset of participants who identified their ethnicity as European Caucasian (EC) to determine if ethnicity had an impact on the relationship between rs2235048 genotype and our endpoints.

## 3. Results

### 3.1. Sample Characteristics

Forty-eight participants were included in the analysis (four of the eligible 52 participants were missing blood cannabinoid data and were excluded). Out of the 48 participants, 10 were rs2235048 TT genotype carriers, 23 were TC carriers, and 15 were CC carriers. The allelic frequencies for rs2235048 were 44.8% T and 55.2% C. The genotypes did not deviate significantly from Hardy-Weinberg Equilibrium in these 48 participants (*p* > 0.1). For reference, according to the NCBI Allele Frequency Aggregator (https://www.ncbi.nlm.nih.gov/snp/rs2235048 Accessed on 15 August 2022), the frequency of the C allele in a general population (*n* = 110,452) is 49.9%. In the EC subgroup, 4 were TT genotype carriers, 14 were TC carriers, and 8 were CC carriers, and the allele frequencies were thus 42.3% T and 57.7% C.

The baseline characteristics of participants are summarized in Table 1. Participants were on average 22.40 years old, with average BMI of 24.77 kg/m^2^, and on average used cannabis 2.55 days per week. A significant difference in frequency of cannabis use was detected across *ABCB1* rs2235048 genotypes, where C-allele carriers had a significantly greater mean weekly frequency of use compared to the TT genotype group (TT = 1.85 ± 0.24, TC/CC = 2.74 ± 0.14 days/week, t(46) = 3.04, *p* = 0.004). There was no significant difference in age, IQ, sex, ethnicity, or BMI across rs2235048 genotypes.

### 3.2. Estimate Dose of THC and Blood Cannabinoids

Estimated dose of THC did not differ significantly between genotype groups (TT = 76.60 ± 6.08, TC/CC = 84.18 ± 3.60, t(46) = 1.0, *p* = 0.33). This was also true in the subset of participants who self-identified as EC (TT = 76.41 ± 3.53, TC/CC = 81.32 ± 5.01, t(24) = 0.41, *p* = 0.69).

Time courses of whole-blood THC, 11-OH-THC, and THC-COOH from baseline to 360 min are presented as a function of genotype in Figure 1. There was no significant genotype by time interaction for THC (F(1.06,45.64) = 0.28, *p* = 0.61), 11-OH-THC (F(1.22,52.38) = 0.53, *p* = 0.50), or THC-COOH (F(1.27,54.46) = 0.28, *p* = 0.65). This remained true in the subset of EC participants.

Pharmacokinetic parameters are presented in Table 2. Maximum concentrations (Cmax) of THC and both metabolites did not differ between genotype groups. C-allele carriers had significantly higher AUC of THC-COOH (TT = 45.92, TC/CC = 95.21, t(40.45) = 2.09, *p* = 0.043) and marginally significantly higher AUC of 11-OH-THC (TT = 3.28, TC/CC = 7.18, t(41.23) = 2.02, *p* = 0.050), though there was no significant difference in AUC of THC. In the subset of EC participants, C-allele carriers had significantly higher Cmax of THC-COOH (TT = 11.9 ± 3.46, TC/CC = 30.54 ± 6.60, t(21.61) = 2.50, *p* = 0.02) and AUC of THC (TT = 14.14 ± 4.29, TC/CC = 31.00 ± 6.11, t(17.03) = 2.26, *p* = 0.037), 11-OH-THC (TT = 2.38 ± 0.97, TC/CC = 8.73 ± 2.21, t(22.81) = 2.63, *p* = 0.015), and THC-COOH (TT = 28.58 ± 7.21, TC/CC = 113.99 ± 26.94, t(22.21) = 3.06, *p* = 0.006).

### 3.3. Subjective Effects

Time courses of VAS ratings of the seven scales (Effect, High, Good Effects, Bad Effects, Liking, Rush, and Like Cannabis) are presented in Figure 2. There was no significant genotype by time interaction for Effect (F(2.51,100.45) = 0.20, *p* = 0.87), High (F(2.56,102.53) = 0.31, *p* = 0.79), Good Effects (F(3.16,126.40) = 0.16, *p* = 0.93), Bad Effects (F(4.11,164.28) = 0.41, *p* = 0.81), Liking (F(3.60,143.99) = 0.90, *p* = 0.46), Rush (F(2.47,98.83) = 0.52, *p* = 0.63), or Like Cannabis (F(3.21,125.30) = 1.09, *p* = 0.36). These results did not change drastically in the subset of EC participants, though in this subset there was a main effect of genotype on VAS Bad Effects (F(1,20) = 5.83, *p* = 0.025), where ratings of Bad Effects were higher in the TT group compared to TC/CC.

Maximum VAS effect and AUC data are presented in Table 3. Overall, there were no significant differences between genotype groups, except for a significantly higher mean VAS Liking AUC in C-allele carriers compared to the TT genotype group (TT = 15,895.56, TC/CC = 35,398.33, t(36.92) = 2.49, *p* = 0.017). In the subset of EC participants, C-allele carriers had lower maximum VAS Bad Effects (TT = 52.75 ± 9.69, TC/CC = 21.62 ± 5.18, t(23) = 2.46, *p* = 0.022) and Like Cannabis (TT = 90.75 ± 5.34, TC/CC = 71.10 ± 7.41, t(16.48) = 2.15, *p* = 0.047).

## 4. Discussion

In this human laboratory study, we sought to determine whether the *ABCB1* rs2235048 polymorphism would influence the pharmacokinetics of THC and self-reported subjective cannabis effects in young adults after smoking a single cannabis cigarette. Specifically, we hypothesized that the rs2235048 C allele would be associated with lower blood THC and metabolite concentrations, as well as lower subjective responses to smoked cannabis. This hypothesis was not supported, as we found no significant genotype difference in blood THC concentrations or subjective effects and found actually higher metabolite concentrations in C-allele carriers compared to the TT genotype.

Our blood cannabinoid data suggested that the rs2235048 C allele is associated with greater concentrations of THC and metabolites in blood after smoking a single cannabis joint. Though only the AUC of 11-OH-THC and THC-COOH were statistically greater in C-allele carriers compared to the TT genotype group, THC and metabolite concentrations were numerically higher in the CC/TC group at every time point. This finding contrasts with the results of Kebir et al. (2018) [16], where lower THC concentrations were reported in T-allele carriers of the rs1045642 polymorphism, which is linked to the C allele of rs2235048. Kebir et al. suggested that the lower blood THC concentrations in T-allele carriers could be due to the rs1045642 T allele (and by linkage, the rs2235048 C allele) leading to lower P-gp expression, thus resulting in less efflux of THC from the central compartment [16]. However, the participants included in their sample had diagnosed DMS-IV cannabis dependence, were consuming an average of 21 cannabis joints per week, and were on average 29.5 years old [16], which is a very different than our sample of young adults using cannabis on average 2.5 days per week who had never met criteria for cannabis dependence. In addition, in our study, cannabis was administered under placebo-controlled laboratory conditions, and thus our blood cannabinoid data were collected at known, fixed time points following cannabis exposure. In contrast, Kebir et al. sampled blood at a variable time after last personal use of cannabis [16]. Thus, while our findings may seem contradictory, it is possible that variation in the *ABCB1* gene impacts THC pharmacokinetics differently depending on the cannabis use history of the population being sampled and on the timing of blood cannabinoid concentration analysis.

In order to account for the potential influence of ethnicity on the relationship between *ABCB1* genotype and THC pharmacokinetics, we performed a secondary analysis in the subset of participants identifying as European Caucasian (EC). In the EC subgroup, the effects of rs1045642 genotype on THC and metabolite concentrations were even more pronounced, with AUC of THC, 11-OH-THC, and THC-COOH and Cmax of THC-COOH statistically significantly higher in C-allele carriers. These findings should be interpreted with caution as there were only 4 participants in the TT group in this subgroup. However, these results suggest that ethnicity is an important variable to consider in future work characterizing the role of *ABCB1* in THC pharmacokinetics.

While our data suggest a potential association between rs2235048 genotype and THC pharmacokinetics, it is important to note that the C-allele carriers in our study did have a significantly greater mean weekly frequency of cannabis use (2.74 days/week, compared to 1.85 days/week in the TT genotype group). In addition, while the difference was not statistically significant, C-allele carriers had a greater estimated dose of THC in the *ad libitum* paradigm than the TT genotype group (84.18 mg compared to 76.60 mg). Thus, in lieu of a pharmacokinetic explanation for the genotype difference in blood cannabinoid concentrations, it is possible that C-allele carriers, on average, typically used cannabis with greater frequency and intensity than the TT group, and so chose to smoke more of the cannabis cigarette during the experimental session.

Across the seven subjective effects scales, we saw minimal evidence of meaningful genotype differences. We did see significantly higher AUC for the “liking” measure in C-allele carriers, but this is likely to be a spurious finding since we did not see a consistent pattern in subjective effects disaggregated by genotype as we have observed in other studies, e.g., associations with *OPRM1* polymorphisms [8]. No previous studies were identified that examined *ABCB1* influences on acute subjective cannabis effects. The most relevant finding is the association between the rs1045642 C allele and greater risk of cannabis dependence [15]. In their follow-up study that found greater THC concentrations in C-allele carriers, Kebir et al. [16] suggested that greater efflux of THC from the central compartment in C-allele carriers could encourage more frequent use of cannabis to maintain a drug high, and could also cause greater severity of cannabis withdrawal symptoms in individuals with a CUD. It could be that the putative association between *ABCB1* polymorphisms and CUD is primarily related to pharmacokinetic, and not pharmacodynamic, differences, which would explain our finding of no genotype differences in self-reported subjective drug effects.

While these preliminary data support a role of *ABCB1* genetic variation in moderating cannabis-related phenotypes, there are a few important limitations to highlight. We were limited by a relatively small sample size. We hope to address this limitation in future work as we recruit larger samples and can replicate and expand the current results (e.g., examine the influence of other *ABCB1* polymorphisms, which could help to clarify inconsistencies between our results and previous findings). We believe our laboratory-based candidate gene approach is an important complement to population-based genetic designs such as GWAS, given our ability to measure a variety of well-described cannabis-related phenotypes such as blood cannabinoid concentrations and subjective effects. Replicating our findings in a larger sample could also allow us to address issues of confounding. In the present sample, we saw a higher mean frequency of cannabis use in C-allele carriers, which could be confounding the genotype difference we saw in blood cannabinoid concentrations. Thus, in a larger sample, we could control for group differences in cannabis use patterns. We were also limited in our capacity to investigate ancestry influences on our genetic association, as we had to dichotomize our ethnicity variable. There are known ethnicity/ancestry differences in rs1045642 allele frequency, e.g., T allele frequency of 56.1% in Caucasians, 20.2% in African Americans, 40% in Asian Americans, and 50% in both Mexican Americans and Pacific Islanders [23]. The study was also limited by the use of rs2253048 as a proxy for rs1045642 (as this SNP was not assayed by the genotyping array). Although these SNPs are in linkage disequilibrium, the correlation between alleles is not 100%. In other words, the rs2235048 C allele may not exactly correspond to rs1045642 T allele. Furthermore, there are other *ABCB1* SNPs in linkage disequilibrium with rs1045642, and thus we cannot be sure of the true causal variant [17].

In conclusion, though our findings were mostly negative, we have provided the first preliminary data from a human laboratory paradigm suggesting an association between *ABCB1* rs2253048 and acute cannabis measures in young adults, where the C allele was associated with some measures of blood cannabinoid concentrations (but no difference in self-reported subjective drug effects) after smoking a single cannabis cigarette. In addition, this relationship appears to be greater in individuals who self-identify a European Caucasian ethnicity, thus ethnicity is an important variable to consider in future research examining relationships between *ABCB1* variation and THC pharmacokinetics. While these data are preliminary and require replication in a larger sample where potential confounds can be controlled, they support a small but growing body of literature that suggests the *ABCB1* gene moderates a variety of cannabis-related phenotypes and may be involved in conferring risk to CUD.

## Figures and Tables

**Figure 1 brainsci-12-01189-f001:**
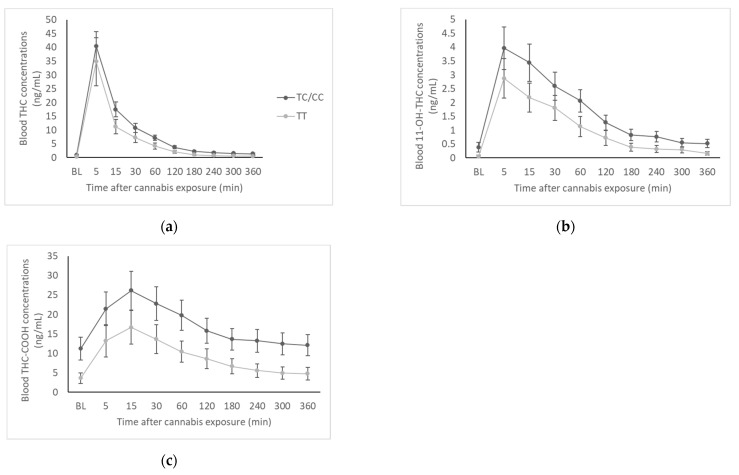
Blood cannabinoid concentrations by genotype group. (**a**) Whole-blood concentrations of THC at baseline and for 6 h (360 min) following exposure to cannabis; (**b**) Whole-blood concentrations of 11-OH-THC at baseline and for 6 h (360 min) following exposure to cannabis; (**c**) Whole-blood concentrations of THC-COOH at baseline and for 6 h (360 min) following exposure to cannabis.

**Figure 2 brainsci-12-01189-f002:**
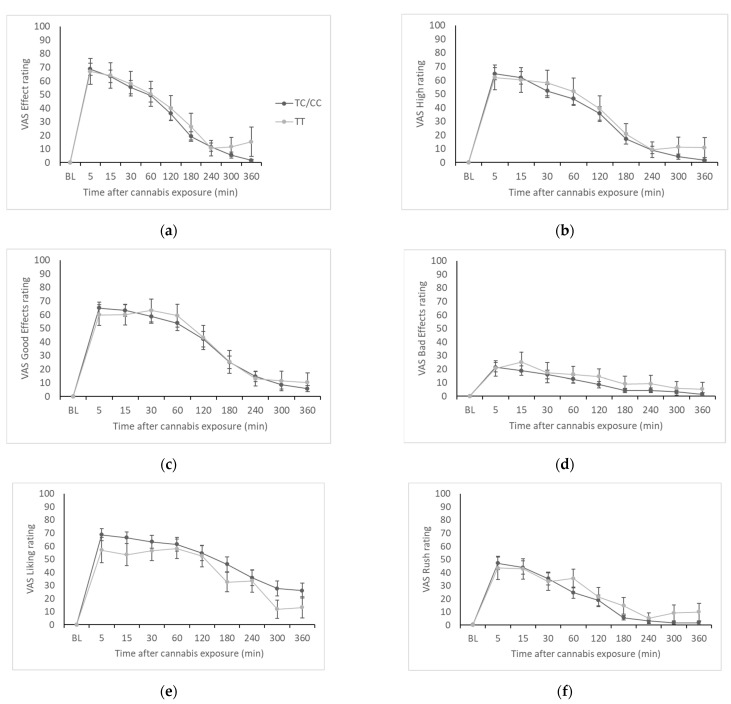
VAS ratings by genotype group. (**a**) VAS Effect; (**b**) VAS High; (**c**) VAS Good Effects; (**d**) VAS Bad Effects; (**e**) VAS Liking; (**f**) VAS Rush; (**g**) VAS Like Cannabis.

**Table 1 brainsci-12-01189-t001:** Sample characteristics.

	Total Sample (*n* = 48)	TC/CC (*n* = 38)	TT (*n* = 10)	*p*
Age (years)	22.40	22.55	21.80	0.26
BMI (kg/m^2^)	24.77	24.88	24.36	0.76
Frequency of cannabis use (days/week)	2.55	**2.74**	**1.85**	**0.004**
Sex (% female)	29%	24%	50%	0.10
Ethnicity (% EC)	54.2%	57.9%	40.0%	0.31

Bolded values are significant at *p* < 0.05; EC, European Caucasian.

**Table 2 brainsci-12-01189-t002:** Pharmacokinetic parameters.

Parameter	TC/CC (*n* = 36)		TT (*n* = 10)		*p*
	Mean	SE	Mean	SE	
THC C_max_	41.20	32.76	34.80	27.77	0.58
THC AUC	28.57	25.49	17.09	13.89	0.18
11-OH-THC C_max_	4.21	4.69	2.87	2.27	0.39
11-OH-THC AUC	*7.18*	*9.64*	*3.28*	*3.40*	*0.050*
THC-COOH C_max_	**26.82**	**29.44**	**17.47**	**14.0**	**0.034**
THC-COOH AUC	**95.21**	**116.12**	**45.92**	**42.38**	**0.043**

Bolded values are statistically significant at *p* < 0.05; italicized values are marginally significant.

**Table 3 brainsci-12-01189-t003:** VAS maximum effect and Area-under-the-Curve (AUC).

Parameter	TC/CC (*n* = 37)		TT (*n* = 10)		*p*
	Mean	SE	Mean	SE	
VAS Effect max	72.16	26.36	68.80	30.62	0.73
VAS Effect AUC	9905.38	8306.62	14,812.50	23,070.26	0.55
VAS High max	68.49	26.11	64.60	28.76	0.68
VAS High AUC	9135.08	7556.54	13,160.28	19,161.07	0.33
VAS Good Effects max	73.40	27.92	67.90	26.83	0.58
VAS Good Effects AUC	13,803.94	14,775.60	15,042.22	20,351.91	0.84
VAS Bad Effects max	27.62	24.90	33.30	25.88	0.53
VAS Bad Effects AUC	3826.82	6727.82	8951.11	14,812.31	0.34
VAS Liking max	76.27	27.65	68.20	27.43	0.42
**VAS Liking AUC**	**35,398.33**	**37,233.72**	**15,895.56**	**13,200.68**	**0.017**
VAS Rush max	52.11	30.90	49.20	27.29	0.79
VAS Rush AUC	5675.00	7941.076	8089.72	12,306.32	0.48
VAS Like Cannabis max	76.11	31.60	73.70	32.68	0.83
VAS Like Cannabis AUC	30,052.11	32,786.97	16,591.11	14,921.72	0.088

Bolded values are statistically significant at *p* < 0.05.

## Data Availability

The data presented in this study are available upon request from the corresponding author. The data are not publicly available due to study participant confidentiality mandates.

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
