# Peer review of "Association between ABCB1 rs2235048 Polymorphism and THC Pharmacokinetics and Subjective Effects following Smoked Cannabis in Young Adults"

_brainsci, 2022, doi:10.3390/brainsci12091189_

Round 1

Reviewer 1 Report

The authors study the relation between rs2235048 polymorphism in cannabis healthy consumers and subjective effect an cannabinoid PK parameter

The study design was correct,  the manuscript could be improved.

Line 77 : "can influence PD and PK"

results from previous studies concerning CNR1 eand OPMR1 genotype must be rapidly evoqued.

L.159 : precise "the masses = "the doses" in the paper

L180 Kebir et al. 2018 = ref 14

3.1 results must be compared from general population

The EC subgroup analysis must be detailled: maybe a table or a simple number particularly in the 3.2, we don't now how many C-carrier's EC participants

It s' hard to correlate the VAS Linking in table 3 results with the VAS linking rating of figure 2e.  The AUC 0-360 calculation must be precised

AUC/Dose could be better than AUC alone without ponderation

The authors must discuss why the rs1045642 was not studied,

the design of the study does not allow the following extrapolation L 342 : we saw a higher mean.... it's only declarative, the sex ratio is too different (24% vs 50%)

Author Response

We thank the reviewer for their helpful comments. Responses to individual comments are below in bold type.

The authors study the relation between rs2235048 polymorphism in cannabis healthy consumers and subjective effect an cannabinoid PK parameter

The study design was correct,  the manuscript could be improved.

Line 77 : "can influence PD and PK"

The purpose of this sentence was to convey that by influencing PK, efflux pumps can have an impact on PD. The sentence has been revised to clarify this meaning.

results from previous studies concerning CNR1 eand OPMR1 genotype must be rapidly evoqued.

We have added a sentence in the introduction to mention our previous studies of CNR1 and OPRM1 genotypes and subjective cannabis effects.

L.159 : precise "the masses = "the doses" in the paper

The masses of the cigarettes are not exactly synonymous with the doses, as the cannabis cigarettes contain components such as the filter and rolling papers that are not pharmacologically relevant. This is why we specify that the masses of the cigarettes were used to estimate the THC dose. We added a sentence here to clarify how estimated THC dose was calculated.

L180 Kebir et al. 2018 = ref 14

Thank you for catching this oversight, the proper in-text citation has been added.

3.1 results must be compared from general population

General population allele frequency has been added to this section.

The EC subgroup analysis must be detailled: maybe a table or a simple number particularly in the 3.2, we don't now how many C-carrier's EC participants

The count of genotype carriers and the allelic frequencies for the EC subgroup have been added to section 3.1

It s' hard to correlate the VAS Linking in table 3 results with the VAS linking rating of figure 2e.  The AUC 0-360 calculation must be precised

The calculation of the AUC is now specified in section 2.5 (Data Analysis).

AUC/Dose could be better than AUC alone without ponderation

We appreciate the suggestion, and agree this would be preferable in some situations. However, as we mentioned earlier in this response, the THC “dose” variable is estimated, not known precisely, so AUC/dose would likely just add more noise to the variable, and would not be preferable in our study.

The authors must discuss why the rs1045642 was not studied,

We have clarified that rs1045642 was not assayed by the genotyping array we used in the limitation paragraph of the discussion.

the design of the study does not allow the following extrapolation L 342 : we saw a higher mean.... it's only declarative, the sex ratio is too different (24% vs 50%)

We disagree with this comment – we are simply here observing that our C-allele carrier group had a higher mean cannabis use frequency than the homozygous TT group. There is no extrapolation, we’re simply commenting on the data we collected. We have made a slight edit to the sentence to clarify that we’re referring to the sample included in our study, not trying to generalize and state that C-allele carriers have greater frequency of cannabis use.

Reviewer 2 Report

It's good.

Congratulations!

Author Response

We thank the reviewer for their support of our manuscript.